# Towards the generation of synchronized and believable non-verbal facial behaviors of a talking virtual agent

Alice Delbosc*†
Davi, The Humanizers
Puteaux, France
alice.delbosc@lis-lab.fr

Magalie Ochs
CNRS, LIS, Aix-Marseille University
Marseille, France
magalie.ochs@lis-lab.fr

Nicolas Sabouret
CNRS, LISN, Paris-Saclay University
Orsay, France
nicolas.sabouret@universite-paris-saclay.fr

Brian Ravenet
CNRS, LISN, Paris-Saclay University
Orsay, France
brian.ravenet@limsi.fr

Stéphane Ayache
CNRS, LIS, Aix-Marseille University
Marseille, France
stephane.ayache@lis-lab.fr

## ABSTRACT

This paper introduces a new model to generate rhythmically relevant non-verbal facial behaviors for virtual agents while they speak. The model demonstrates perceived performance comparable to behaviors directly extracted from the data and replayed on a virtual agent, in terms of synchronization with speech and believability. Interestingly, we found that training the model with two different sets of data, instead of one, did not necessarily improve its performance. The expressiveness of the people in the dataset and the shooting conditions are key elements. We also show that employing an adversarial model, in which fabricated fake examples are introduced during the training phase, increases the perception of synchronization with speech. A collection of videos demonstrating the results and code can be accessed at: https://github.com/aldelb/non_verbal_facial_animation.

## CCS CONCEPTS

• **Computing methodologies** → **Neural networks**; **Animation**.

## KEYWORDS

Non-verbal behavior, behavior generation, embodied conversational agent, neural networks, adversarial learning, encoder-decoder

**ACM Reference Format:**
Alice Delbosc, Magalie Ochs, Nicolas Sabouret, Brian Ravenet, and Stéphane Ayache. 2023. Towards the generation of synchronized and believable non-verbal facial behaviors of a talking virtual agent. In *INTERNATIONAL CONFERENCE ON MULTIMODAL INTERACTION (ICMI '23 Companion), October 9–13, 2023, Paris, France.* ACM, New York, NY, USA, 10 pages. https://doi.org/10.1145/3610661.3616547

---

*Also with CNRS, LIS, Aix-Marseille University.
†Also with CNRS, LISN, Paris-Saclay University.

---

## 1 INTRODUCTION

Interest in virtual agents has grown in the last few years, their applications are multiplying in games or virtual environments, for instance in the medical domain [1, 50]. However, these virtual agents are not yet widely used in practice, partly because of their lack of natural interaction, which discourages user engagement [5].

In order to address this issue, Cassell [8] propose to integrate various natural modalities of human behavior into the virtual agent, including speech, facial expressions, hand gestures, body gestures, and more. Facial expressions, gestures, and gaze direction are examples of non-verbal behavior, encompassing actions distinct from speech. While psychologists argue about the percentage of information non-verbally exchanged during an interaction [77], it is clear that the non-verbal channel plays an important role in understanding human behavior.

Several studies show that facial expressions, gaze direction, and head movements are essential non-verbal behaviors that play a crucial role in conveying a speaker's intentions and emotional state [49], and could even improve the way a virtual agent is perceived in general [6, 42]. Munhall et al. [46] also showed that the rhythmic beat of head movements increases speech intelligibility. In the same way, Tinwell et al. [63] showed that "uncanniness" is increased for a character with a perceived lack of facial expressions. In this paper, we present a machine-learning based model to generate non-verbal facial behaviors that take into account facial expressions, head movements and gaze direction.

The process of generating non-verbal behavior for a specific speech can be approached from different angles, such as generating natural and believable behaviors, generating behaviors that are rhythmically synchronized with the speech, adapted to the intonation or appropriate to the semantic content of the speech. In this work, we chose to focus on generating rhythmically relevant and believable non-verbal behaviors for the virtual agent as he speaks. This involves creating a framework that generates non-verbal features that align with the rhythmic patterns of speech.

The paper is organized as follows. We provide an overview of existing works in section 2, followed by the formulation of the learning problem in section 3. After presenting the datasets used and their processing methodologies in section 4, we describe the used architecture in section 5. In section 6 we present our research

question and hypotheses. The section 7 is dedicated to our evaluation methods and results. Finally, we conclude the paper and introduce perspectives in section 8.

## 2 RELATED WORK

The research works on behavior generation can be characterized by various aspects, such as the adopted approach (rule-based or data-driven), the dataset characteristics, the inputs, and outputs of the model, and more. To provide a structured overview of the state-of-the-art, we organize it as follows: in section 2.1, we present examples of rule-based models; in section 2.2, we describe data-driven models including deep learning models; in section 2.3 we present the different possible input for the models and their impact on the generated behaviors; and in section 2.4, we discuss the output's representation of the models.

### 2.1 Rule-based approaches

The first approaches explored for the automatic generation of virtual characters' behavior were based on sets of rules. The rules describe the mapping of words or speech features to a facial expression or gesture. One of the first works to explore the latent relationship between speech and gesture to generate realistic animation was Cassell et al. [9] with *Animated Conversation*. Kopp and Wachsmuth [33] proposed a model-based approach for generating complex multimodal utterances (*i.e.*, speech and gesture) from XML specifications.

The development of new rule-based systems often required the development of a new domain-specific language (DSL). These DSLs were often incompatible with each other, even if the systems solved similar or overlapping goals [49]. A group of researchers developed a unified language for generating multimodal behaviors for virtual agents, called behavior Markup Language (BML). BML has become the standard format for rule-based systems, and many other works have followed using this format [43, 56].

It is important to point out that these approaches focused on intention. They were highly effective in terms of communication, but not very natural, since they mainly inserted predefined animations [49]. More recent research has therefore begun to explore data-driven systems.

### 2.2 Data-driven approaches

Data-driven approaches do not depend on experts in animation and linguistics. They learn the relationships between speech and movements or facial expressions from data. They are born out of proof of a strong correlation between an individual's speech and her/his non-verbal behavior [31, 44]. For example, Yehia et al. [70] and Honda [27] show that pitch is correlated with head motions.

Marrioryad and Busso [42] proposed to replace rules with Dynamic Bayesian Networks (DBN). In Chiu and Marsella [10], a Gaussian Process Latent Variable Model (GPLVM) has been used to learn a low-dimensional layer and select the most likely movements given the speech as input. Recently, Yang et al. [69] proposed a motion graph-based statistical system that generates gestures and other body movements for dyadic conversations. Hidden Markov Models (HMM) were used to select the most likely body motion [39, 43] or head motion [60] based on speech. However, these research works are still based on an animation dictionary, limiting the

diversity of the generated movements. Moreover, in these models, there is only one motion sequence for an input audio signal. It supports the hypothesis that the speech-to-motion correspondence is injective, but the correspondence between acoustic speech features and non-verbal behavior is a "One-To-Many" problem [38].

More recently, deep neural networks have demonstrated their superiority in learning from large datasets by generating a sequence of features for non-verbal behavior. The main objectives of these deep learning-based systems are the naturalness and the synchronization between audio and speech. For example, Kucherenko et al. [35] proposed an encoder-decoder speech to motion. However, the traditional deterministic generative models employed in this approach often suffer from a limitation: they tend to generate average motion representations [36]. To address this limitation, researchers have explored the integration of probabilistic components into their generative models. Notably, popular probabilistic models such as Generative Adversarial Networks (GANs) [18, 58, 62], Variational Autoencoders (VAEs) [21, 24, 40], and diffusion models [12, 13, 73] have been employed.

GANs [20] can be used to convert acoustic speech features into non-verbal behaviors while preserving the diversity and multiple nature of the generated non-verbal behavior. However, GANs are reputed to be unstable and suffer from a specific problem called the collapse mode. The collapse mode is a very common failure that causes the model to generate only one behavior. Numerous of work has been done to improve their training [2, 45].

In comparison with rule-based approaches, data-driven approaches have made advancements in terms of naturalness. The generated behaviors played on virtual agent have shown a perceived naturalness that, in certain work, surpasses the actual behaviors. However, several limitations persist, particularly concerning the perceived appropriateness of these behaviors in relation to the accompanying speech, still quite far away from the ground truth [38].

Moreover, even though numerous gestural properties can still be inferred from speech, the generated behaviors will unavoidably overlap with the audio and text channels [37]. That implies that data-driven approaches are significantly less communicative than rule-based approaches.

New architecture has recently begun to combine these two approaches, in an attempt to take advantage of the benefits of both while minimizing the drawbacks. For example, the work of Zhuang et al. [76] uses a transformer-based encoder-decoder for face animation and a motion graph retrieval module for body animation. Another example is the work of Ferstl et al. [19], who generates parameters such as acceleration or velocity of motion from the audio, before finding a corresponding motion in a database.

As we chose to focus on the generation of behaviors that are rhythmically coherent and believable, regardless of semantic appropriateness, we chose a data-driven approach. Given the performance of GANs in the area of non-verbal behavior generation, we implemented an adversarial model, more precisely a Wasserstein Generative Adversarial Network (WGAN).

### 2.3 Inputs of the models

Inputs to motion generation models can take the form of audio input [26, 34], textual input [4, 72], or both [17, 71].

Kucherenko et al. [37] showed that text and audio differ in their use, the time-aligned text helps predict gesture semantics, and prosodic audio features help predict gesture phase. Early deep learning systems ignored semantics, taking only audio inputs. These approaches using only audio can produce well-synchronized movements, which correspond to rhythmic gestures, but the absence of text transcription implies that they will lack structure and context, such as semantic meaning [49]. More recent approaches attempt to integrate semantics to generate meaningful gestures, taking as input text or audio and text.

Other forms of input are used, such as non-linguistic modalities (e.g. interlocutor behavior) [28, 48] or control input (e.g. style parameters transmitted during model inference) [17, 23]. The ability to control body motion based on a specific input signal, such as their emotional state or a social attitude, can significantly improve the usability of the method [23].

Since our objective in this work is limited to generating behaviors that are as rhythmically coherent and credible as possible, we will only use audio as input for our model.

## 2.4 Outputs of the models

Speech-driven facial animation is a process that automatically synthesizes speaking characters based on speech signals. The majority of work in this field, such as those presented above, creates a correspondence between audio and behavioral features. Then the behavioral features are used to animate a character, such as the *Greta* virtual agent. This is the approach we use in our work.

Other systems directly generate face images, often from real individuals, without relying on behavioral features. We are not going to discuss these models in detail, as their outputs are very different from ours. However, we note that the architectures employed are relatively similar. Vougioukas et al. [65], Zhou et al. [75] used a temporal GAN, and Kim and Ganapathi [32] used a VAE model.

Works that generate behavioral features can generate them in various categories. Many studies focus on the automatic generation of body movements [13, 74]. Most head- and/or face-based methods generate either facial animations or head movements exclusively. The generation of facial expressions and head movements poses distinct challenges: head movements exhibit greater diversity across individuals compared to facial expressions. However, it is important to acknowledge that facial expressions and head movements are inherently interconnected and synchronized with speech [9]. Habibie et al. [25] or Delbosc et al. [11] introduced an adversarial approach for the automatic generation of facial expressions and head movements jointly. Drawing inspiration from these works, our research focuses on analyzing facial expressions and head movements in a combined manner, with a representation of facial expressions using explainable features, specifically facial action units.

Body and head movements are generated using 3D coordinates, ensuring uniformity in their representation. However, the generation of facial expressions offers a range of approaches. They can be generated directly with the 3D coordinates of the face, like Karras et al. [30] or describe using a model, such as FLAME model [41] in Jonell et al. [28], FaceWarehouse model [7] in Pham et al. [54] or Basel Face ModelPaysan et al. [52].

Our task requires a representation that allows the encoding of the facial gestures from video, the simulation of them on a virtual agent, and the possibility to manipulate the generated facial expressions. Therefore, we represent the facial expressions using action units (AUs) based on the well-known Facial Action Coding System (FACS)[15]. Thanks to this representation, we can use *Openface* to extract the facial expression from videos, play our generated facial expression on the *Greta* platform and, in the future, adapt the generated action units to express particular social attitude [14, 64] (see section 8). This is why we consider it particularly important to represent facial expressions with AUs.

## 3 PROBLEM FORMULATION

Our task can be formulated as follows: given a sequence of acoustic speech features $F_{speech}[0:T]$ extracted from a specific segment of speech input at regular intervals $t$, the task is to generate the sequence of corresponding behavior $\theta_{behavior}[0:T]$ that a virtual agent should perform while speaking.

The sequence $\theta_{behavior}[0:T]$ consists of three components: $\theta_{head}[0:T]$, $\theta_{gaze}[0:T]$, and $\theta_{AU}[0:T]$, representing head movements, gaze orientation, and facial expressions, respectively. The head movements $\theta_{head}[0:T]$ and gaze orientation $\theta_{gaze}[0:T]$ are specified using 3D coordinates, while the facial expressions $\theta_{AU}[0:T]$ are defined using action units (AUs) based on the Facial Action Coding System (FACS) [15]. These notations will be consistently employed throughout this article.

After generating the behaviors, we evaluate them with both objective and subjective evaluations. To simulate the generated behaviors on a virtual agent, we use the *Greta* platform [53]. This process of generating and evaluating the behaviors is visually described in figure 1.

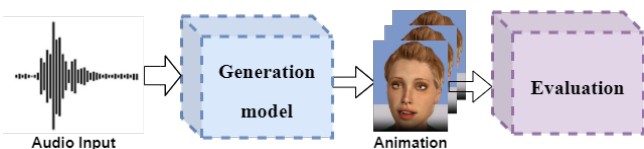

**Figure 1: The generation and evaluation process**

Compared to the state of the art, the contributions of this work are: (1) a new adversarial model for speech-driven non-verbal facial behavior generation, with facial behavior generation based on action units; (2) a comparison between using a small amount of suitable data and a larger amount of data (adding less suitable data), to train our model; (3) an evaluation of the effects of adding new relevant fake fabricated examples during the training phase of the adversarial model.

## 4 FACIAL BEHAVIORS DATASETS

Among the main challenges linked to the generation of non-verbal behaviors, the research community frequently highlights some issues. In particular, the difficulty of finding suitable training data.

Various methods, which differ in terms of cost and time requirements, are available for data collection. On one hand, there are expensive and time-consuming approaches, such as employing multiple cameras and motion capture systems. On the other hand, faster but less precise methods involve simple recording techniques

combined with tools designed to extract the desired features directly from videos. Even if datasets exist, they may be small, not contain the required features, their quality can be insufficient, etc.

In our specific task, we require dataset that emphasize facial recordings. For anticipating our future work, we also need them to contain interaction scenarios and various social attitudes. Without a doubt, behaviors generated through data-driven approaches will inevitably be constrained by the data on which they are trained. For instance, when it comes to generating behaviors based on a particular social attitude, the ability to generate "angry" behavior will rely on the presence of such behavior in the initial dataset. We utilize two datasets in our research.

### 4.1 Selected datasets

We utilize the *Trueness* dataset [51], a newly created multimodal corpus containing 18 interaction scenes on discrimination awareness in a forum theater. All interactions are in French. We chose it for several reasons. Firstly, it contains scenes of interaction, simulating conflicts, played by actors with different social attitudes (denial, aggressive, conciliatory). What's more, the scenes are shot by actors who make sure they stay in the camera's field of view, so the camera only films the face and torso.

For a larger amount of data, we employ additionally the *Cheese* dataset [55], selecting 10 interaction scenes involving free conversation of students, in French. We chose this dataset because it also contains interaction scenes. The difference with *Trueness* is that these aren't actors, and they aren't conflict scenes, so their behavior is less expressive. This dataset also differs in terms of shooting conditions, the students are located a little further away from the camera and almost their entire bodies are filmed.

For both dataset, each video is divided into two parts, representing the perspectives of the first and second persons of the interaction. We obtain approximately 3h40 of recording time for *Trueness* and approximately 5h of recording time for *Cheese*. As these are interaction scenes, we've made sure that both parts of the same interaction belong to the same subset (train set or test set).

We aim to investigate the impact of incorporating the second dataset during training on the model's performance. By utilizing both datasets, the model will have access to a larger volume of training data. However, the *Trueness* dataset contains more expressive facial expressions, and the actors are filmed at closer angles. Consequently, throughout this article, we will refer to the other dataset, *Cheese*, as having "farther-away shooting conditions" and being "less expressive".

To integrate these data sets into our models, we automatically extract behavioral features and acoustic speech features from the existing videos using state-of-the-art tools, namely *Openface* [3] and *OpenSmile* [16].

### 4.2 Features extraction and processing

*Openface* is a toolkit that detects automatically the head position, gaze orientation, and facial action units of a person on a video. Features are extracted at the frequency of 25 frames per second (25 fps). We consider the eye gaze position represented in world coordinates, the eye gaze direction in radians, the head rotation

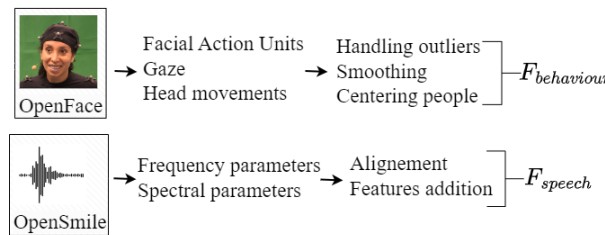

**Figure 2: Extraction and processing of data**

in radians, and 17 facial action units in intensity from 1 to 5[1]. We obtain a total of 28 features characterizing the head, gaze, and facial movements. These features, noted $\theta_{behavior} \in \mathbf{R}^{28}$, are used for the training and constitute the output of the generation model.

*OpenSmile* is a toolbox that extracts the eGeMAPS audio features from speech. This tool extracts features at a frequency of 50 fps. To eliminate redundancy between acoustic speech features, we conducted a correlation study, and we finally kept seven spectral and frequency parameters[2]. For each of them, the first and second derivatives are computed [25, 68]. We also add a binary feature that indicates whether the person is speaking or not (i.e. 1 for "speaking" and 0 for "listening"). In total, we consider 22 audio features. The audio features extracted from the human speech are noted $F_{speech} \in \mathbf{R}^{22}$.

To ensure that our model learns from clean and plausible data, we need to remove the frames that *Openface* has incorrectly processed. For example, frames where faces are obscured by a hand or hair, or where excessive head movements are done. Thanks to the visual analysis of a few behaviors extracted with *Openface* and directly replayed on *Greta*, we identify outliers and the treatment required:

- identification and deletion of outliers frames;
- creation of transitions if frames have been deleted;
- smoothing of features with a median filter with a window size of 7 to eliminate *Openface* noises;
- centering of the head and gaze coordinates so that the virtual agent faces the user;
- alignment of acoustic speech and behavioral features at 25 fps.

Finally, to enhance the model's understanding of speaking and listening behaviors and improve behavior synchronization with speech, we set the coordinates of the head and gaze, and the intensity of the AUs at a constant when the protagonist is not speaking. These adjustments highlight the distinction between "speaking" and "listening" behaviors.

The most widely used method for the generation of human behavior consists in working on short segments over a sliding window varying from a few seconds to several minutes depending on the socio-emotional phenomena studied [47]. Inspired by this method, the videos in the dataset were cut into segments of 4 seconds.

---

[1] AU01, AU02, AU04, AU05, AU06, AU07, AU09, AU10, AU12, AU14, AU15, AU17, AU20, AU23, AU25, AU26, AU45.
[2] alphaRatio, hammarbergIndex, mfcc1, mfcc2, mfcc3, F0semitoneFrom27.5Hz, logRelF0-H1-H2.

# 5 FACIAL BEHAVIOR GENERATION MODEL

Following the research conducted during the state of the art, our proposed model[3] adopts an adversarial encoder-decoder framework. It generates head movements and gaze as 3D coordinates and facial expressions as AUs intensities. Unlike certain works [35, 59], no smoothing is applied to the output.

To minimize the generation of highly improbable behaviors, we employ a normalization step at the input of our models, coupled with sigmoid activation layers in the model's output. The normalization scales the input data between 0 and 1 and the sigmoid layer constrains the generated data to fall within the range of values observed in our training data. As a result, the generated data should closely resemble real data while still allowing the generation of novel patterns.

As we adopt an adversarial approach, the model work as a game between two networks: a generator and a discriminator. While the discriminator is optimized to recognize whether an input is generated by the generator or taken from the real data, the generator tries to fool the discriminator by learning how to generate data that looks like real data. Figure 3 illustrates our model.

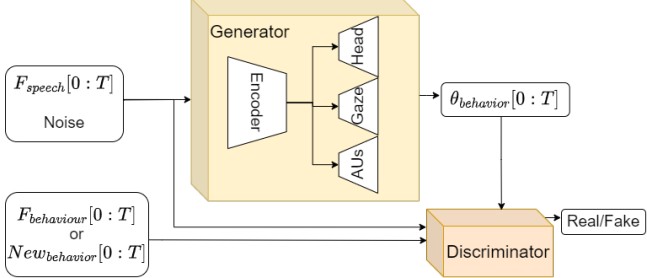

**Figure 3: The overall architecture of our model**

## 5.1 The generator

The generator generates data by sampling from a noise distribution $Z$ and acoustic speech features $F_{speech}[0...T]$. The noise enables to keep the randomness of the generated movements. To generate our noise, we generate two random digits, and use these two values to create a noise of size 200, with transition digits that progressively follow between the first and second digits. This allows us to create a gradual evolution from the first digit to the second, ensuring a certain cohesion in the noise generated for each sequence.

The generator takes the form of a 1D encoder-decoder. It is an adaptation of the U-Net implementation [57] originally created for 2D image segmentation. The encoder starts by learning a representation of the acoustic speech features, then concatenates it with the noise. It consists of five blocks, which we call *DoubleConv*. The *DoubleConv* block is constituted of convolution 1D, batch normalization 1D, and Relu, and those twice. The convolutional layers have kernels of size 3 and dropout after each of them. The last 4 blocks are followed by MaxPool. Then, three decoders are created to generate believable behaviors.

[3]https://github.com/aldelb/non_verbal_facial_animation.

Each decoder is associated with a data type with different value intervals: a decoder for head movements, a decoder for eye movements, and a decoder for AUs. They consist of four *DoubleConv* blocks and UpSampling after the first 4 blocks. As the decoders are symmetric with the encoder, it uses skip-connectivity with the corresponding layers of the encoder. They end with a sigmoid activation layer. Figure 4 illustrates this architecture.

We supervise our generator $G$ with the following loss function:

$$\mathcal{L}_G = \mathcal{L}_{gaze} + \mathcal{L}_{head} + \mathcal{L}_{AU}$$

$\mathcal{L}_{gaze}$, $\mathcal{L}_{head}$ and $\mathcal{L}_{AU}$ are the root mean square errors (RMSEs) of the gaze orientation, head movement, and AUs features.

$$\mathcal{L}_{gaze} = \sum_{t=0}^{T-1} (\theta_{gaze}[t] - \hat{\theta}_{gaze}[t])^2$$

$$\mathcal{L}_{head} = \sum_{t=0}^{T-1} (\theta_{head}[t] - \hat{\theta}_{head}[t])^2$$

$$\mathcal{L}_{AU} = \sum_{t=0}^{T-1} (\theta_{AU}[t] - \hat{\theta}_{AU}[t])^2$$

## 5.2 The discriminator

The generator receives real examples from real data and fake examples generated by the generator. Both the generator and the discriminator receive acoustic speech features $F_{speech}[0...T]$. The discriminator can thus measure if the behavior looks natural, but above all if the behavior looks natural with respect to these acoustic speech features, and if the temporal alignment is respected.

An important aspect of our architecture is that the discriminator does not only receive real and fake generated examples. We create a new data type, called $New_{behavior}$. $New_{behavior}$ are fake examples designed to facilitate the learning of synchronization between speech and behaviors. These examples associate acoustic speech features of a "speaking" person with behavior features of a "listening" person (and vice versa).

The discriminator starts by learning a representation of the acoustic speech features and a representation of the behavioral features. After concatenating these two representations, there are four *DoubleConv* blocks and MaxPool after each block. We add dropout after the convolutional layers. It ends with a linear layer and a sigmoid activation layer. Figure 4 illustrates this architecture.

## 5.3 Training details

We choose to implement a Wasserstein GAN [2] and, more specifically, a Wasserstein GAN with gradient penalty [22]. GANs try to replicate a probability distribution, this implementation uses a loss function that reflects the distance between the distribution of the data generated and the distribution of the real data.
We pose the adversarial loss function with the discriminator $D$:

$$L_{adv}(G, D) = \mathbb{E}_{F_{speech}}[D(F_{speech}, G(Z, F_{speech})]$$

$$-\mathbb{E}_{F_{speech}, \theta_{behavior}}[D(F_{speech}, \theta_{behavior})] + \lambda \underset{\hat{x} \sim \mathbb{P}_{\hat{x}}}{\mathbb{E}}[(||\nabla_{\hat{x}}D(\hat{x})||_2 - 1)^2]$$

The point $\hat{x}$, used to calculate the gradient norm, is any point sampled between the distributions of the generated data and the

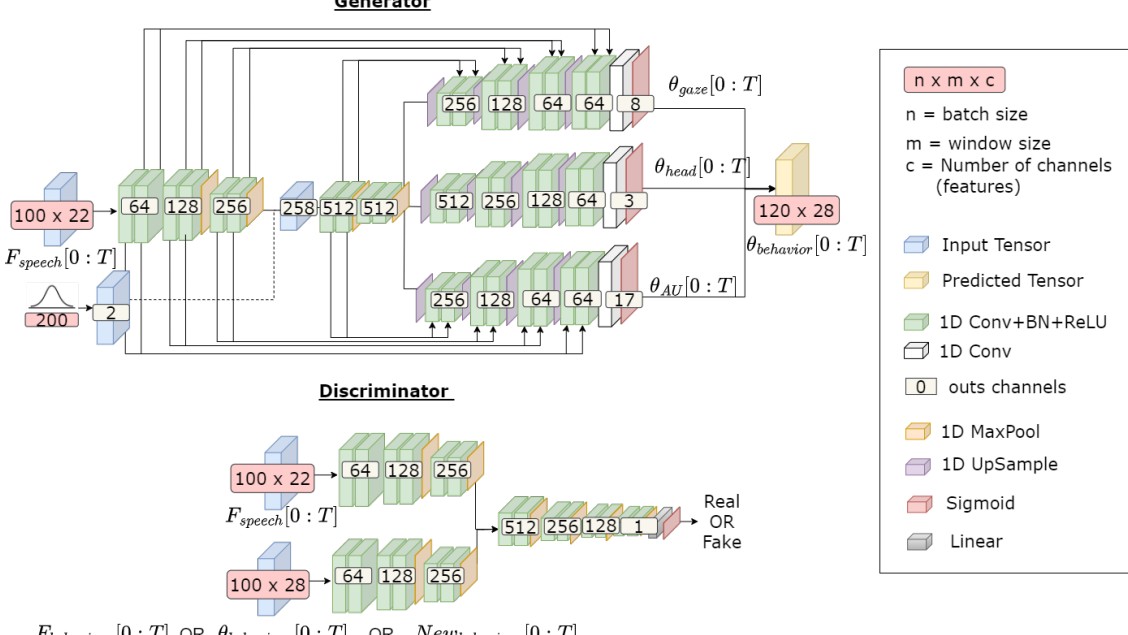

**Figure 4: The detailed architecture**

real data $\hat{x} = t\theta_{behavior} + (1 - t)F_{behavior}$ with $0 \leq t \leq 1$.
As the original paper, we use $\lambda = 10$

We use Adam for training, with a learning rate of $10^{-4}$ for the generator and $10^{-5}$ for the discriminator. Our batch is size 32.

Combining the adversarial loss with the direct supervisory loss, our objective is :

$$\mathcal{L} = \mathcal{L}_G + w.\mathcal{L}_{adv}(G, D)$$

With w set to 0.1 to ensure that each term is equally weighted.

Based on this architecture, we would like to analyze two important aspects: the data considered during model training and the addition of fake examples $New_{behavior}$.

## 6 RESEARCH QUESTIONS AND HYPOTHESES

We want to know which factors influence the model to obtain more or less human-like behaviors and speech-matched behaviors. We make the following assumptions:

H1 The perception of speech/behavior synchronization will be improved with the addition of our $New_{behavior}$ examples during training (section 5.2).

H2 The addition of *Cheese* during training, will improve the perception of believability.

H3 The addition of *Cheese* during training, will degrade the perception of synchronization.

Our intuition behind the last two hypotheses is that the actors' dataset *Trueness* is more distant from everyday behavior than the "less expressive" dataset *Cheese*. On the other hand, the "farther-away shooting conditions" dataset *Cheese* is less suited to the generation of facial behaviors (section 4.1). Based on our hypotheses, we will compare the following models:

m1 architecture presented in section 5, trained on *Trueness* dataset.

m2 m1 model with the association of *Trueness* and *Cheese* datasets for training.

m3 model m1 without our fake examples $New_{behavior}$, during model training.

GTS "Ground Truth Simulated" are the extracted behavior from the data, directly simulated on the virtual agent. We use the term "simulated" because the resulting videos are not exactly a replication of the human's behavior, due to the limitation of *Openface* and *Greta* (limited number of AUs for example).

Videos of each condition can be found on *YouTube*[4].

## 7 EVALUATION

We evaluate our models through both objective and subjective methods. Objective evaluations are quantitative metrics, while subjective evaluation is done through user-perceptive study.

Objective metrics are often inappropriate [38] and always insufficient when it comes to comparing different architectures in behavior generation. These metrics fail to capture the coherence between behaviors and speech, as they primarily focus on statistical similarity to recorded motion rather than contextual appropriateness. Subjective evaluations play a crucial role in assessing the complexity of social communication. However, conducting subjective studies can be time-consuming and complex, which is why objective metrics are employed to complement the evaluation process.

Comparing results across different behavior-generation studies is challenging due to the lack of a standardized baseline in the field. Different works often rely on disparate data sources for training

---

[4]https://www.youtube.com/playlist?list=PLRyxHB7gYN-Cs127qTMJIR78fsQu_8tZQ

their models, the features generated differ according to their objectives, and the visual representations of the generated gestures also vary with different avatars and software, thereby influencing the perception of the generated behavior [67].

It is therefore important to note that, due to the unique nature of our task, especially the differences in output compared to previous research works, direct performance comparisons with existing models are not applicable. For the time being, comparison with behaviors extracted and replayed directly on virtual agents enables us to get around some of these issues, we refer to it here as *ground truth* or *ground truth simulated* (*GTS*).

## 7.1   Objective evaluation

The objective measures are based on algorithmic approaches and return quantitative values reflecting the performance of the model. We consider comparisons of distributions and measurements of acceleration and jerk.

**Dynamic Time Warping:** DTW is used to compare the distance between *ground truth* distributions and generated distributions. We measure the distance for each generated feature and present the results averaged over all features. The distribution closest to the *ground truth* distribution is the one with the lowest value.

**Table 1: Distance between GTS and generated distributions –** *Average score (mean) and standard deviation (std).*

|  | $m1$ | | $m2$ | | $m3$ | |
|---|---|---|---|---|---|---|
|  | mean | std | mean | std | mean | std |
| DTW | **451.23** | 11.08 | 484.57 | 11.55 | 460.90 | 12.11 |

**Average acceleration and jerk:** The second derivative of the position is called acceleration, and the third time derivative of the position is called jerk. It is commonly used to quantify motion smoothness [38]. A natural system should have average acceleration and jerk very similar to the ground truth. We calculate these two metrics for the first eye, the second eye, the head, and present the results averaged over all of these features.

**Table 2: Acceleration (Acc.) and jerk –** *Average score (mean) and standard deviation (std).*

|  | $GTS$ | | $m1$ | | $m2$ | | $m3$ | |
|---|---|---|---|---|---|---|---|---|
|  | mean | std | mean | std | mean | std | mean | std |
| Acc. | 10.71 | 0.79 | 13.49 | 1.38 | **9.10** | 0.42 | 19.02 | 1.89 |
| Jerk | 458.48 | 47.49 | **545.66** | 52.04 | 358.62 | 16.59 | 768.00 | 58.91 |

These metrics were evaluated for all the videos with *Trueness* test set. Tables 1 and 2 show the results, the closest numbers from the simulated ground truth are bold.

In terms of acceleration and jerk. We note that $m2$ is smoother than *GTS*. According to our hypotheses, the perceived believability of the smoother model must be the best. For the distance between the generated distributions and the ground truth distribution, the $m1$ model is the closest. The perception of the synchronization of

the model with the closest distribution to the ground truth must be superior to others.

If objective metrics provide valuable insights, they have limitations and are not sufficient to assess the complexity of social communication. We need to conduct subjective studies to confirm or refute our hypotheses (section 6).

## 7.2   Subjective evaluation

The subjective measures are based on the evaluation of human observers. To select the appropriate evaluation criteria, we base our subjective evaluation study on previous research [38, 66]. We evaluate two criteria through direct questions:

  o  believability: how human-like do the behaviors appear?
  o  temporal coordination: how well does the agent's behavior match the speech? (In terms of rhythm and intonation)

We randomly selected four videos from our *Trueness* test set, two with female voices and two with male voices. This selection allowed us to demonstrate the flexibility of our models in generating non-verbal behaviors for different virtual agents on *Greta*.

Following the recommendation of Wolfert et al. [66], we opted for a rating-based evaluation. In this method, participants assign ratings to the generated behaviors in all conditions (*GTS*, $m1$, $m2$, $m3$). Ratings rather than pairwise comparisons are recommended when more than 3 conditions are under consideration, pairwise comparisons tend to become unwieldy for 4 or more conditions.

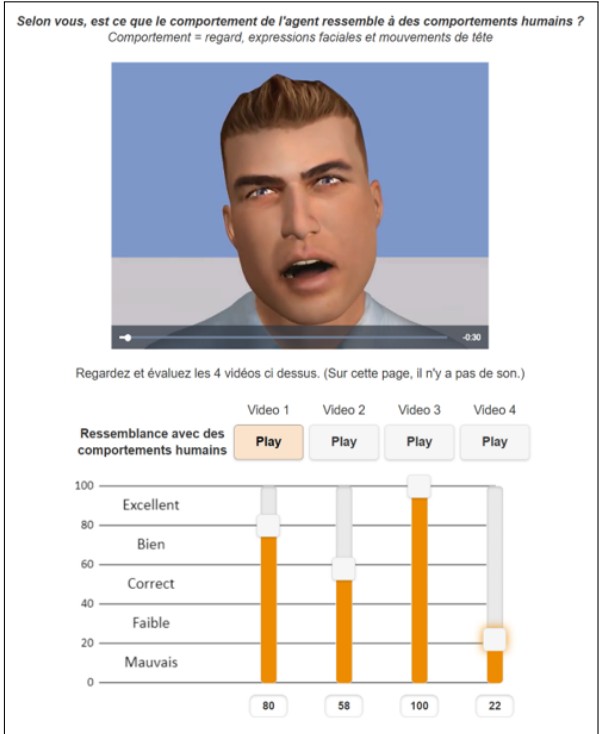

**Figure 5: Interface of our subjective evaluation tool**

To create the study, we developed an interface inspired by the works of Jonell et al. [29] and Schoeffler et al. [61]. Through several

videos, we ask participants to rate the behaviors of several virtual agents in terms of believability and coordination. We specified that when we talked about behaviors, we referred to facial expressions, head movements, and gaze.

We divided the evaluation of each criterion into separate parts with specific instruction pages. First, the evaluation of believability, in which the videos were silent, they did not include any audio. This allows videos to be rated only with the behaviors performed and not on their relationship to the speech. Secondly, the evaluation of temporal coordination, for which the videos are presented with sound corresponding to the virtual agent's behaviors.

Figure 5 provides an example of one of the pages used in the evaluation process. On each page, at the top and in bold, a question is displayed, corresponding to the criterion being rated. Participants must watch 4 videos on the page (corresponding to each condition) and rate each video using the scales. The scale is from 0 (worst) to 100 (best) and can be set by adjusting a slider for each video. On any given page, the videos can be viewed as many times as they like, but they can't go back to the previous page.

By considering the two evaluated criteria, the four selected sequences of videos, and the four conditions, we obtained a total of 32 videos to rate, each approximately of 30 seconds in duration. The whole evaluation takes about 20 minutes.

Thirty persons, with a good level of French, recruited on social networks, participated in our study (16 males and 14 females). The average age of the participants is 28 years, with a standard deviation of 8.06. They viewed each of the videos, in a random order, and rated them on each of the criteria. Table 3 presents the results of this subjective evaluation for our three selected models and $GTS$.

**Table 3: Results of the perceptive study** – *Average score (mean) and standard deviation (std) for both coordination (Coo.) and Believability (Bel.) on all 4 conditions.*

|      | $GTS$ | | $m1$ | | $m2$ | | $m3$ | |
|------|------|------|--------|-------|--------|-------|--------|-------|
|      | mean | std | mean | std | mean | std | mean | std |
| Coo. | 36.53 | 19.67 | **43.42** | 19.04 | 38.82 | 18.69 | 38.77 | 20.91 |
| Bel. | 47.60 | 17.33 | 45.39 | 14.81 | **58.74** | 15.48 | 39.02 | 16.17 |

Statistical analysis is conducted to assess significant differences between the models. First, the normality of the data is assessed using the Shapiro-Wilk test, which indicates that the data are from a normally distributed population. Therefore, a repeated measures ANOVA is performed.

The results reveal the superiority of $m1$ compared to $m3$ in terms of synchronization ($p < .05$) and also in terms of believability ($p < .01$). Our first hypothesis is significantly validated, and the addition of our fake example during the training of our adversarial model improves the perception of speech/behavior synchronization.

We can also observe the dominance of $m2$ in terms of believability compare to m1 ($p < .01$) but the superiority of $m1$ in terms of coordination ($p < .05$). Hypotheses two and three are also significantly validated. The addition of "less expressive" and "farther-away shooting conditions" data increases the perception of believability, but reduces the perception of synchronization.

Another interesting result is the comparison between $m1$ and $GTS$. The differences are not significant, but $m1$ tends to outperform $GTS$ in terms of synchronization ($p = .067$), an uncommon result in the field of behavior generation. We hypothesize that setting "listening" behaviors to 0 and adding our fake examples greatly improves the perception of synchronization with speech.

## 8 DISCUSSION AND FUTURE WORK

We presented a new approach for the generation of rhythmically coherent behavior during the speech of a virtual agent. Our model demonstrates perceived performance comparable to behaviors extracted from data and replayed on a virtual agent, in terms of synchronization with speech and believability. This approach, based on an adversarial model, is enriched with fake examples of our own creation and trained on one or two datasets.

We found that adding data during the training, doesn't necessarily increase performance. The expressiveness of people within the dataset and shooting conditions are key elements. The addition of these data during training generates smoother movements, increasing the perceived believability of the generated behaviors but reducing the perception of synchronization with speech.

The fake examples provided to the model reduce the distance between the distributions of generated data and *ground truth* data, enhancing the perception of synchronization and believability of generated behaviors.

These results should be interpreted cautiously, especially due to potential influences of our non-verbal behavior extraction and visualization tools on participant perception in subjective evaluation. Given the subjective evaluation duration and complexity, we only tested 4 randomly chosen sequences, unlikely to represent the full dataset. Moreover, participant numbers might not unveil all notable differences between conditions, particularly comparing *simulated ground truth* and our model.

This work is part of a larger project to generate socio-affective non-verbal behaviors during social interaction training. Several perspectives are therefore on the horizon. After the generation of rhythmically coherent behavior during speech, we aim to generate semantically and contextually relevant non-verbal behaviors for the virtual agent during speech. This entails associating specific behaviors with the semantic content of the agent's speech. By aligning non-verbal behaviors with the intended meaning of the agent's utterances, we will enhance the communicative effectiveness and expressiveness of the virtual agent.

To incorporate the socio-affective dimension and be able to simulate different types of scenarios, we will introduce a constraint in the generation process, focusing on a particular social attitude. This step involves encoding the desired social attitude (aggressiveness, consilience, or denial), and using it to guide the generation of non-verbal behaviors.

After that, we will take into account the signals and behaviors exhibited by the human interlocutor. This will enable the virtual agent to dynamically adjust its non-verbal behavior to match and engage with the interlocutor.

Studies on behavior generation opens the way to agents capable of generating expressive behaviors from speech. A great opportunity in the field of training, where they can reproduce believable situations in a safe environment, while ensuring user engagement.

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
