# OpenReview forum: "Towards the generation of synchronized and believable non-verbal facial behaviors of a talking virtual agent"
_ACM.org/ICMI/2023/Workshop/GENEA — GENEA Workshop 2023_

### Official Review · Reviewer_VxJb · 2023-08-06
**Well-written and conducted work**

**Rating:** 8
**Confidence:** 3

**Review:**

The paper is generally well-written and referenced. In particular, the related work provided a good overview of the topic and factors involved in designing a solution for the problem of generating believable facial and head motion for artificial agents. The study design also provided many relevant references to support the approach. The statistics conducted as part of the study are well presented and justified. I appreciate that the authors provide both objective and subjective metrics for the work. The paper provides a nice contribution to a growing and valuable area of research.

The effort towards reproducibility is strong. The code is made publicly available on github, the data is made available, and the study materials are made available. While I have not tried to run the code, from browsing the repository, the code looks easy to follow and the documentation is comprehensive. I would suggest adding a license to the repository so the community can know the usage terms.

Possible improvements:

One area where I feel that the paper is lacking is in a clear statement of contribution to the field. Including a description of the intended novel elements of this work toward the end of Section 2, or in Section 3 would help the reader to understand the differences between the work put forth here and those from the existing literature. Helping the reader to understand why the approach adopted here might be better than existing approaches is particularly useful when there is no comparison to an alternative model baseline.

The study hypotheses would benefit from some additional reasoning or references to support them. In particular, H2 suggests that 'less expressive' data is more believable. However, it seems like an assumption that the non-acted emotions are less expressive than the acted ones between the two datasets - this indeed seems likely, but I am wondering if there is any data to support this assumption? Additionally, as the output is conditioned on the input, it seems possible that the more expressive emotions occur with certain audio input, and if this type of audio is not part of the test set, those more expressive animations would not be produced anyway. It is unclear from which underlying dataset the 4 test videos were from - if they match (or not) to the respective model, this may bias the results (i.e., if the 4 videos are from the cheese dataset, it seems clear that m2 would have an advantage over m1, as m2 has seen other data from that dataset and m1 has not). Adding this detail to the paper would be good.

Finally, it seems that there may be some limitation in the underlying tools or techniques used in extracting and playing back the animations, indicated by the relatively low perceptual score for both coordination and believability of the simulated ground truth. It would be interesting to compare with the original video to understand more about whether this originates from the extraction/playback process, or if these randomly selected videos might have other issues.

**Nominate For A Reproducibility Award:**

Yes, code, data and study materials publicly available.

---

### Official Review · Reviewer_d6pp · 2023-08-08
**The authors present a WGAN-based approach to generate facial animation from audio features only as a substep of a larger project**

**Rating:** 7
**Confidence:** 3

**Review:**

The paper is well written and easy to read.
It seems like a good contribution to GENEA given that the authors focus on one specific problem and provide valuable insights about it, namely about how to treat training data for a model that can generate facial expression from audio features, in a way that actually reduces the amount of necessary data.
This has clearly been stated as one substep of a larger project so I'm also interested in seeing more in the future.

There is obviously still a lot of work to do.
I watched the videos and my first comment would be - why use Greta which is a very outdated software?
It looks very primitive and blank compared to contemporary computer graphics.
It also seems like its limited animation and texturing capabilities may shadow the benefits and positive findings of your method.

Maybe related to this - or not - the SGD which are supposed to be a (fake) representation of the ground-truth, do not seem to match the audio recordings that much either. I mean in the sense that they should be considered a baseline, they fail quite a lit to even represent jaw opening which is generally the most basic "lip-sync" motion.
I'm currently looking at sgd - scene 2 in particular right now.

I find it hard to judge any of the SGD versions as remotely believable so while a subjective comparative evaluation may be valid (and your methodology in general seems well designed), I find it dubious to conclude whether the result is actually usable for a real-world scenario or not.
This can be commented on the paper, or can be further discussed at the workshop:
- How valueable is this evaluation, if the tool isn't really up to what you are trying to evaluate?
- Is the approximated baseline case actually an acceptable representation of the real baseline?
Given this is just an initial step of a larger problem, we can tolerate it for now, but I hope the authors may consider this comment in their future work.

Second, in all m1 m2 and m3 videos, the results is still quite jittery which really break the believability to me.
Unless they were meant to represent very agitated, stressed characters (which I understand is not the case).
Is there a way to reduce the jitteriiness of the resulting signal?
- Can the authors identify or perform an educated guess about what is the part of their method/architecture that is to blame for jittery motion?

Given my comments, I think the authors should present their work at the GENEA workshop to receive feedback that can be valueable for their future steps.

---

### Decision · Program_Chairs · 2023-08-11

**Decision:**

Accept

**Comment:**

As both reviewers recommended accepting this paper with high scores, the chairs decided to accept this paper as a workshop paper. Please read the reviews carefully and update the paper for the camera-ready version.

Specifically,
* GDS looks not believable enough even if it is from human motion. Further discussion on this and clarification of the current limitation would be helpful. (And GDS and SGD terms are mixed in the paper)
* Lack of support for the hypotheses.